# KCTD Proteins Have Redundant Functions in Controlling Cellular Growth

**DOI:** 10.3390/ijms25094993

**Published:** 2024-05-03

**Authors:** Robert Rizk, Dominic Devost, Darlaine Pétrin, Terence E. Hébert

**Affiliations:** Department of Pharmacology and Therapeutics, McGill University, 3655 Promenade Sir-William-Osler, Room 1303, Montréal, QC H3G 1Y6, Canada; robert.rizk2@mail.mcgill.ca (R.R.); dominic.devost@mcgill.ca (D.D.); darlaine.petrin@mcgill.ca (D.P.)

**Keywords:** KCTD proteins, Heterotrimeric G proteins, cell growth, cellular signaling

## Abstract

We explored the functional redundancy of three structurally related KCTD (Potassium Channel Tetramerization Domain) proteins, KCTD2, KCTD5, and KCTD17, by progressively knocking them out in HEK 293 cells using CRISPR/Cas9 genome editing. After validating the knockout, we assessed the effects of progressive knockout on cell growth and gene expression. We noted that the progressive effects of knockout of KCTD isoforms on cell growth were most pervasive when all three isoforms were deleted, suggesting some functions were conserved between them. This was also reflected in progressive changes in gene expression. Our previous work indicated that Gβ1 was involved in the transcriptional control of gene expression, so we compared the gene expression patterns between GNB1 and KCTD KO. Knockout of GNB1 led to numerous changes in the expression levels of other G protein subunit genes, while knockout of KCTD isoforms had the opposite effect, presumably because of their role in regulating levels of Gβ1. Our work demonstrates a unique relationship between KCTD proteins and Gβ1 and a global role for this subfamily of KCTD proteins in maintaining the ability of cells to survive and proliferate.

## 1. Introduction

The KCTD (Potassium Channel Tetramerization Domain) family consists of 25 proteins that show a significant amount of structural diversity beyond conserved *Bric-à-brac*, *Tramtrack*, and *Broad complex* (BTB) domains, which are responsible for the oligomerization of KCTD proteins. The functions of all these proteins impact a number of cellular processes in health and disease but remain incompletely understood (reviewed in [1,2,3,4,5]), although they serve many roles in regulating the ubiquitination of other proteins (reviewed in [3]). KCTD proteins can be subdivided, based on homology, into several subgroups, including the related KCTD2, 5, and 17 (Figure 1A). Although these three KCTD proteins have all been shown to have common binding partners, such as Gβγ subunits [6,7,8], whether they have both unique and conserved functions remains an active area of investigation. It has become clear that KCTD proteins play major roles in regulating GPCR signaling [8,9,10,11]. It has been demonstrated using proteomic approaches that multiple isoforms of Gβγ pairs interact with several KCTD proteins [7]. In fact, all three proteins can sensitize Gβγ-mediated modulation of adenylyl cyclase activity, suggesting they have a conserved function as Gβγ-interacting proteins [9,11]. However, it remains unclear why the control of signaling events by KCTD proteins varies from cell type to cell type. In fact, the effects of the KCTD proteins, in many ways, have been shown to be cell type-specific, suggesting that they are regulating different actors in different cells. The published studies of the phenotypic effects of KCTD proteins [3,8,9,10,11,12,13] demonstrate a remarkable plasticity, which is likely due to different levels or types of KCTD proteins in the different cell models used.

We and others have also shown that Gβ1 is ubiquitinated or associated with ubiquitinated partners ([14] see also [15,16,17])**,** suggesting KCTD proteins regulate the levels of Gβ subunits, which would have dramatic effects on signaling networks regulated by Gβγ subunits. If KCTD proteins are centrally involved in controlling the expression levels of distinct populations of Gβγ subunits, this would go a long way toward explaining the pleiotropic effects seen when KTCD levels are changed, as Gβγ subunits are central players in the organization and regulation of cellular signaling. Some residual ubiquitination remains when KCTD5 is knocked out, and this is sensitive to MG-132 [14], suggesting the other homologs might also be involved. Here, we focus on the effects of knocking out KCTD2,5, and 17 to assess their effects on cell phenotypes in HEK 293 cells. We demonstrate a functional redundancy between the three isoforms, which impacts cell proliferation. We also demonstrate that knocking out GNB1 exhibits contrasting effects compared to the knockout of KCTD isoforms regarding the modulation of G protein expression.

**Figure 1 ijms-25-04993-f001:**
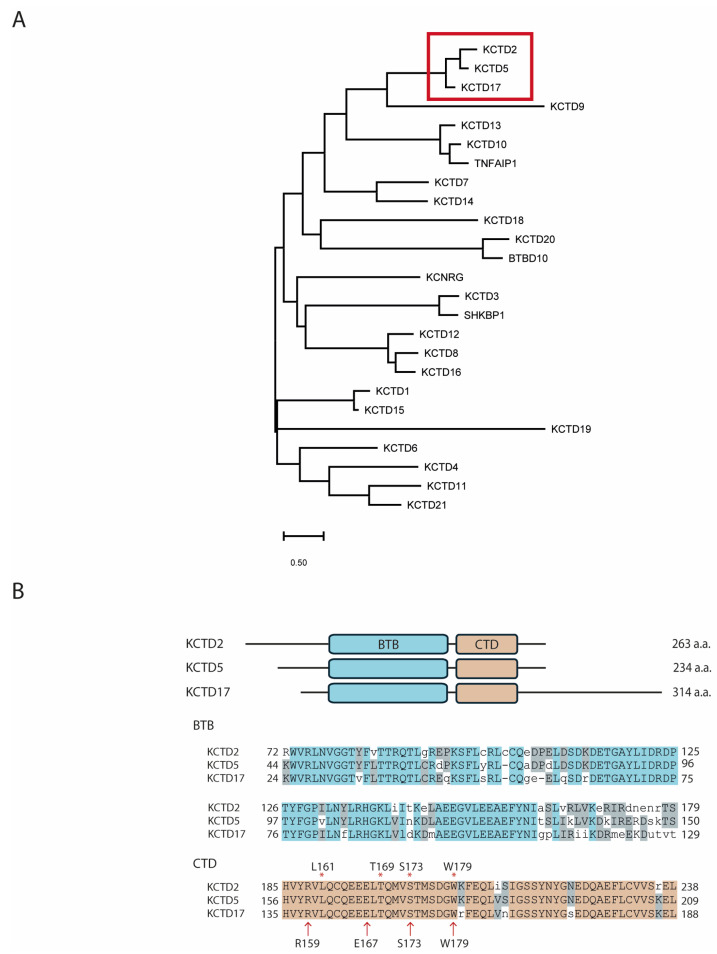
Evolutionary conservation, structure, and sequence similarity among KCTD2, KCTD5, and KCTD17. (**A**) Phylogenetic tree of all 25 human homologs of the KCTD family. Red box shows all three members clustered in the same group. (**B**) **Top**. Schematic representation of human KCTD2, KCTD5, and KCTD17 structural organization of their BTB and CTD domains. **Bottom**. Amino acid sequences alignment of their BTB and CTD domains. Asterisks and arrows refer to key residues of the contact patch between KCTD5 and Gβ1 protein, as described [14,17], respectively.

## 2. Results

The BTB and C-terminal domains of KCTD2, KCTD5, and KCTD17 are well aligned structurally (Figure 1B), indicating that they likely share some redundant functions. To investigate the role of Gβ1 and KCTD2/5/17 in regulating cellular behaviors, their corresponding genes were knocked out using CRISPR-Cas9 in HEK 293 cells. The general strategy was to design efficient and specific sgRNAs that were as close as possible to the translation start site and, ideally, overlapping a restriction site for genotyping convenience with the exception of GNB1 KO, where clone screening was performed by using a Surveyor assay kit (Figure 2A). The DNA from all positive cell clones in RFLP-based screening was sequenced by inactivating corresponding restriction sites or leading to DNA digestion in the Surveyor test. From the sequence analysis, cell clones that carried a translation coding shift and/or resulted in the creation of a premature stop codon in all coding alleles and the concomitant absence of the WT allele were selected (Figure 2B). Validation of the GNB1 KO cell line by western blot clearly showed that the expression of the Gβ1 protein was greatly reduced when compared to parental cells (Appendix A), supporting the idea that the gene editing strategy targeting the GNB1 gene was specific to that G protein subtype; unfortunately, such analysis was not possible for the KCTD-KO cell lines as no specific and sensitive antibodies are available, even after trying several commercial sources. However, we show by RFLP analysis and sequencing that both alleles of the relevant KCTD lines were edited to result in nonfunctional versions of these genes (Appendix A).

To assess the redundancy of these three KCTD isoforms, sequential knockout of these genes was followed by an assessment of their rates of growth in culture. To quantify these changes, CRISPR/Cas9-generated ∆KCTD5, ∆KCTD2/5, and ∆KCTD2/5/17 HEK 293 cells were cultured and subjected to live cell imaging over a 54-h period. Analysis of confluency changes revealed a significant decrease in cell growth over time, reflected by a distinct reduction in slope in sequential KCTD-KO cell lines compared to the parental cells (Figure 3). A noticeable decrease in cell growth rate was observed as presumed redundancy was lost.

### 2.1. Transcriptional Profiles Associated with the Decrease in Cell Growth KCTD-KO Cells

To more comprehensively explore the observed growth phenotypes, we conducted RNA-seq analysis on both KCTD-KO, GNB1-KO, and parental cell lines. Gene expression analysis was performed using Deseq2, and differential expression was filtered based on *p*-adjusted values (*p* < 0.05). Two biological replicates were performed and analyzed (Appendix A). The data are available at the Gene Expression Omnibus (series record GSE262642).

A Gene Ontology enrichment analysis was conducted using the ClusterProfiler package in BioConductor v. 3.18. The EnrichKEGG function identified 294 out of the 5423 differentially expressed genes in crucial proliferation pathways. These pathways encompass mitogen-activated protein kinase (MAPK), Mammalian Target of Rapamycin (mTOR), Phosphoinositide 3-Kinase (PI3K), and epidermal growth factor receptor (ERBB) signaling. Furthermore, these genes exhibited associations with pathways related to cell cycle, apoptosis, and cellular senescence, as depicted in Figure 4.

The ComplexHeatmap package in BioConductor v. 3.18 with the pheatmap function was used to generate heatmaps to visualize differential gene expression across key proliferation pathways in KCTD-KO lines. Employing the Ward D2 method for hierarchical clustering on the selected set of 294 genes related to proliferation pathways (Appendix A), our analysis revealed rather limited changes in gene expression between the ∆KCTD5 and the parental line, again suggesting that there is a certain redundancy when KCTD2 or KCTD17 remain present. In contrast, more pronounced alterations were observed when comparing the parental line to the ∆KCTD2/5 and ∆KCTD2/5/17 lines (Figure 4A). Moreover, 38 genes were found to have log2fold change ≥ 1.5 and a base mean exceeding 100. A heatmap representation of these genes exhibited a clustering pattern reminiscent of the overall gene expression for the 294 genes, emphasizing a marked shift in gene expression in the double and triple KCTD KO lines compared to the parental line. Notably, several differentially expressed genes were identified, including growth factors such as *PDGFA*, *PDGFB*, *RELN*, *TGFA*, and *NTRK2*. These genes play pivotal roles in pathways regulating cell growth and survival (Figure 4B). Future studies will resolve the impact of changes in growth-related or apoptosis-related genes.

### 2.2. Impact of KCTD and GNB1 Knockouts on G Protein Transcriptional Profiles

We established Gβ1 as a regulator of gene expression in cardiac fibroblasts, where it interacts with RNA polymerase II [18]. We also demonstrated that knocking down GNB1 causes an upregulation of Gβ4 protein levels in HEK 293 cells [19]. Gβγ signaling has been associated with a number of transcription regulators [20] (reviewed in [21,22]), and our recent findings highlight its involvement in the regulation of fibrotic gene expression. Specifically, we demonstrated that Gβ1 acts as a brake on the fibrotic response, exerting control over RNA polymerase II activity by destabilizing the elongation complex and negatively influencing fibrotic genes [18]. Gβ1-containing Gβγ subunits may have a unique role in regulating gene expression in cells, helping to set the cellular signaling landscape.

Since KCTD5 subunits regulate levels of Gβ1 [17], we wanted to assess the relative effects on the G protein landscape when either GNB1 or the different KCTD isoforms were knocked out. Given this context, we sought to elucidate the impact of GNB1 and KCTD5 knockouts on G protein gene expression.

Intriguingly, our interest in KCTD5 stems from our prior identification of its role in regulating Gβγ levels through the ubiquitin-proteasome system [15]. Understanding the regulatory influence of KCTD5 on Gβγ prompted us to explore how its knockout would manifest in the broader context of G protein gene expression. To visually capture the expression dynamics, we employed normalized scaled gene counts to construct a heatmap, utilizing the ward d2 method for hierarchical clustering. To address batch effects observed between experiments, two different control conditions were utilized. Consequently, in the heat map representation, values across all four experimental conditions (including the two controls) were scaled uniformly. While this normalization strategy facilitated comparison within each experiment, it is essential to recognize that direct comparisons between the parental line in one experiment and its counterpart in another may yield varying results due to inherent differences in experimental conditions and biological variability.

Notably, our analysis reveals distinct patterns in that knocking out GNB1 results in increased expression levels of most other G proteins compared to the parental line (Figure 5A), in line with our previously observed increases in Gβ4 levels [19], while KCTD5 knockout drove gene expression in the opposite direction, leading to decreased expression of G proteins compared independently to the parental line (Figure 5A, right panels), concordant with observations that KCTD5 knockdown increases the levels of Gβ1 protein in the cell. This exploration not only expands our understanding of G protein regulation but also sheds light on the intricate interplay between KCTD5 and Gβγ within the broader landscape of gene expression. We next evaluated the sequential knockout of KCTD isoforms and noted that there were additive changes associated with knocking out KCTD2 and KCTD17 (Figure 5B), but a clear pattern did not emerge, as sequential knockout also changes growth phenotypes, which are a confound in determining the direct additional effects on Gβ1 levels. The nuanced roles of each individual KCTD isoform on Gβγ signaling remain to be disentangled from effects on other cellular pathways regulated by KCTD isoforms.

### 2.3. Calcium Signalling and KCTD Isoforms

We previously demonstrated that Ca^2+^ signaling is differentially modulated by different isoforms of Gβγ subunits. Through an RNAi screen [19], we uncovered the regulatory role of Gβ4γ1 subunits in mediating carbachol-stimulated calcium release. Additionally, we identified a non-canonical function for Gβ1, showing its involvement in binding to promoter regions of various genes, including GNB4, and enhancing their transcription or translation. Consequently, our interest shifted to understanding the interplay between knocking out different KCTD isoforms and the effects on calcium signaling. In line with the complex global effects, we saw, when knocking out KCTD genes, a seemingly paradoxical increase in calcium signaling despite negative global changes in G protein expression (Figure 6). This is likely due to multiple effects on other proteins beyond changes in the levels of Gβ1 or Gβ4 alone, which may alter calcium signaling in a way more complicated than simply altering levels of a single G protein subunit.

## 3. Discussion

We started with HEK 293 cell lines where KCTD5, or related KCTD2 and KCTD17 [6,7], were knocked out alone or in combination (as these three isoforms have sequence homology suggesting redundant function). Some residual ubiquitination remains when KCTD5 is knocked out, and this is sensitive to MG-132 [14], suggesting the other homologs might also be involved. Loss of KCTD5 increases the levels of Gβγ subunits, and this was rescued by replacing the WT KCTD5 but not KTCD5 ∆C-tail (KCTD5-1-208), which lacks the Gβγ binding site or two KCTD5 point mutants (F128A and L161R) identified as contacts in our Cryo-EM structure [14].

There is significant evidence now that GNB1 serves a more expansive role than the related genes GNB2-4. Recent studies have shown that heterozygous germline mutations in GNB1 (in more than 100 patients to date), the gene encoding the Gβ1 subunit, lead to developmental delay, autism-like symptoms, seizures, and hypotonia [23,24,25,26]. Such mutations are found in several important Gα, Gγ, and effector binding surfaces of Gβ subunits but have not been systematically tested. Loss of Gβ1 in transgenic mice is embryonic lethal [27]. We will focus on germline point mutations in GNB1, with a view toward understanding the impact of these mutations on Gβγ function either in HEK 293 cells or in iPSC-cardiomyocytes and iPSC-fibroblasts. It is likely that disease-causing GNB1 mutations confer gain or loss of function that impacts their interactions with signaling partners and their effects on the signaling landscape in different cell types. We have shown that Gβγ subunits also act as transcriptional regulators, interacting with RNA polymerase II in native HEK 293 cells and primary rat cardiac fibroblasts [18]. In that study, we identified a novel, negative regulatory role for the Gβ1γ dimer in the fibrotic response. Depletion of Gβ1 using siRNA led to de-repression of the fibrotic response at the mRNA and protein levels under basal conditions and an enhanced fibrotic response after sustained stimulation of the angiotensin II type I receptor. We used genome-wide chromatin immunoprecipitation experiments to show that Gβ1 co-localized and interacted with RNA polymerase II on fibrotic genes in an angiotensin II-dependent manner. These results identified Gβ1γ as a novel transcriptional regulator of the fibrotic response, expanding the role for Gβγ signaling, which may have broad implications for the role(s) of nuclear Gβγ signaling in other cell types. Gβγ subunits drive every step of GPCR signaling. Further, we have generated two lines using CRISPR where GNB1 has been knocked out, as well as a comparator line where GNB4 is knocked out. Our RNA-seq experiment on the GNB1-KO line suggests that Gβ1-containing Gβγ subunits play a critical role in regulating gene expression of other G proteins, which suggests that they are critical determinants of the cellular signaling landscape in a given cell, in addition to their more established roles in either cellular signaling per se. Gβγ subunits play key signaling roles in all cells, and studies using general inhibitors have shown promise as drug targets for indications such as heart failure and breast cancer. However, given 60 possible Gβγ combinations in most mammalian cells, it has not been possible to date to target individual pairs. To a large extent, this is because (1) we have only recently appreciated the breadth of their general roles in the cell and (2) we have ignored this complexity, treating them as a single, eponymous “Gβγ subunit”, inhibiting them with “pan” inhibitors such as gallein or βARK-CT. Thus, the cellular population of Gβγ subunits at any time is likely an important determinant of how GPCR signaling complexes form and function. Over the years, we have developed several tools to study the assembly of GPCR signaling complexes. We have shown that Gβγ subunits play roles beyond the simpler actions as signaling molecules by initially interacting with Gα subunits, GPCRs, and effectors such as adenylyl cyclase and Kir3 channels in the ER or Golgi at the latest [28,29,30]. Taken together, our data suggests that receptor, effector, and Gβγ subunits encounter each other very early on during their biosynthesis.

The fact that KCTD2/5/17 knockouts have opposing effects on G protein gene expression compared with knocking out GNB1 suggests that the role these proteins play in Gβ1-mediated events is to regulate their protein levels in the cell. The interplay between the particular compliment of KCTD proteins and Gβγ subunits in cells may go a long way toward explaining the pleiotropic effects of KCTD proteins on cellular signalling in different cell types [3,8,9,10,11,12,13]. Thus, they may be critical regulators of the cellular signaling landscape as well. Beyond this, it is clear that these three KCTD isoforms play additional roles that are so important that some redundancy needs to be built into their shared functions, which would need to be explored through the generation of single KTCD2 and KCTD17 knockout lines. Further studies are required to assess redundancy directly but substituting each of the individual KCTD isoforms in the background of individual or double knockout lines. An additional complexity is the possible effects of heteromeric interactions of distinct KCTD isoforms [31]. Analyses of the distinct KCTD isoforms via structural approaches [32] or with AlphaFold [33,34] may reveal the extent of such events. Future work will also determine if KCTD proteins are implicated in such Gβγ-dependent signaling events in other cell types- but these interactions have already been shown in primary striatal neurons in addition to HEK 293 cells [9].

## 4. Materials and Methods

### 4.1. Sequence Alignment of KCTD Proteins

Amino acid sequences of all 25 human (Homo sapiens) homologs of the KCTD family were collected from the RefSeq: NCBI Reference Sequence Database. Sequences were aligned using ClustalW Multiple Sequence Alignment tool (https://www.genome.jp/tools-bin/clustalw accessed on 1 March 2024). A phylogenetic tree was generated using the Maximum Likelihood Tree method and JTT matrix-based model [35] in MEGA11: Molecular Evolutionary Genetics Analysis version 11 [36]. The tree with the highest log likelihood (−21,549.29) is shown. The tree is drawn to scale, with branch lengths measured in the number of substitutions per site. There were a total of 1050 positions in the final dataset.

### 4.2. Generation of Knockout Lines

The procedure to generate the KCTD5 KO cell line D2 has already been described [14]. The KCTD2 KO was generated in the D2 cell background and then followed by the KCTD17 KO in the newly generated KCTD2/5 KO (2H6) background in a similar approach. In brief, D2 cells were grown in 6 well plates and were transfected using Lipofectamine 2000 with the vector eSpCas9(1.1), (Addgene #71814, Addgene.org) containing the corresponding KCTD2 sgRNA by annealing and phosphorylating these two oligonucleotides: forward oligonucleotide CACCGGCGGAACTGCAGCTGGACC and reverse oligonucleotide AAACGGTCCAGCTGCAGTTCCGCC which were subsequently inserted into the BbsI site of the vector. Correct insertion of the sgRNA was confirmed by Sanger sequencing. For both KCTD2 and 17 (see below), the guide RNA design was performed manually, and the putative off-target sites were quantified using the online software Cas-OFFinder (http://www.rgenome.net/cas-offinder v. 2.4 (accessed several times over the course of the last two years). Forty-eight hours later, cells were detached with trypsin, counted, diluted in fresh media to 1 cell/well/50 μL, distributed into a 96-well plate, and left to grow until confluency. All the wells were then examined under a microscope, and empty wells or wells that apparently contained more than 1 colony were excluded from subsequent generation of clonal lines. Cells were detached and seeded into 6 well plates for expansion as well as into 24 well plates for genotyping, and in both cases, they were left to grow until confluency. Genomic DNA from the cells plated onto the 24 well plates was prepared using the genomic DNA mini extraction kit (Geneaid, FroggaBio, Concord, ON, Canada), and genotyping of each clone was performed using RFLP as a PvuII restriction site was located in the guide RNA sequence. First, a PCR product encompassing the edited region was generated using the forward primer, GACTGCACGAGACACGGCTTG, and the reverse primer, GAGGAGCGTGTCTCAGGACTC, and Q5 high-fidelity DNA polymerase supplemented with the high GC enhancer (NEB). From that screen, the PCR fragment from clone #28 (2H6) was shorter and resistant to PvuII digestion, indicating gene modification. The PCR product was cloned into the cloning vector pMiniT2.0 (NEB) for amplicon sequencing, inserted into NEB10 bacteria by chemical transformation, and plated onto agar plates containing 100 μg/mL ampicillin. Bacterial colonies were grown in LB media in the presence of the same antibiotic, DNA was extracted using a DNA mini-prep kit (Geneaid), and DNA samples sequenced. All colonies analyzed showed a 123 bp deletion encompassing the 5′ UTR, the Met start site, and a portion of the coding exon. The clone was expended and frozen until the next stage of gene editing (to generate the triple knockout line).

The double KCTD2/5 KO cell line 2H6 was transfected with eSpCas9(1.1) containing the specific KCTD17 sgRNA obtained by annealing and phosphorylating the 2 oligonucleotides: CACCGGGAAGAGCTGCAGTCGGAC and AAACGTCCGACTGCAGCTCTTCCC. Genotyping was again performed using RFLP analysis with both PCR forward primers GCGCCCGGGAGGAGGATGCAGAC and the reverse primers CGTTGCCACGGCAATGGGTACATC in conjunction with the restriction enzyme RsrII. From the genotyping screen, all colonies of clone 2B1 showed an extra G in the coding region.

The generation of the GNB1 KO cell line in HEK 293 cells was performed essentially as described, except that genotyping of clones was performed using a Surveyor assay (Integrated DNA Technologies (IDT) cat #70602, Coralville, IA, USA) as no restriction site is present in the editing target sequence. In brief, HEK 293 cells were transfected with the recombinant eSpCas9(1.1) containing the specific GNB1 sgRNA, designed using the online CRISPR Design Tool (http://tools.genome-engineering.org, which no longer exists, unfortunately) and obtained by annealing the 2 oligonucleotides: CACCGCTTGACCAGTTACGGCAGG and AAACCCTGCCGTAACTGGTCAAGC. Twenty-four hours post-transfection, cells were cloned by dilution and surviving cell clones were expended, and levels of edited alleles were determined using the Surveyor kit (IDT) according to the manufacturer recommendations from a PCR product generated with the forward primers: ACCACGCCCACC TTAAACTT and reverse ACATCCTGGGCCAACTAAGA. Finally, the positive clones were sequenced as already described, but from a larger PCR amplicon obtained using the forward primer TGTAATGAGACCATAGTTCTC and the reverse TCAGAAGCTGTAGATTCCAAT C. From this screening, the clone B10 has one allele with a 85 bp insertion and another with a shorter insertion of 45 bp.

### 4.3. Immunodetection of Gβ1

Cells from the parental and GNB1 KO cell lines used to control for editing specificity were grown in 6 well-plated and lysed in RIPA buffer (1% NP-40, 50 mM Tris-HCl pH 7.4, 150 mM NaCl, 1 mM EDTA, 1 mM EGTA, 0.1% SDS, 0.5% sodium deoxycholate) containing protease inhibitors (Sigma, St. Louis, MO, USA) and left on ice for 60 min followed by a brief sonication step also on ice. The lysates were clarified by centrifugation at 15,000 RPM for 5 min at 4 °C, and the supernatants were transferred into a new tube. The protein concentration was evaluated using the BCA kit (Thermo Scientific, Waltham, MA, USA), and the proteins were denatured in Laemmli buffer (50 mM Tris-HCl (pH 6.8), 2% sodium dodecyl sulfate, 10% glycerol, and 0.1 M β-mercaptoethanol) and boiled for 5 min. Moreover, 100 μg of proteins were resolved on a 10% SDS-PAGE, and proteins were transferred onto a PVDF membrane. The same membrane was blocked in TBST (50 mM TRIS pH 7.4, 138 mM NaCl, and 0.1% *vol*/*vol* Tween 20) containing 3% *w*/*v* dry milk for 1 h at RT followed by an overnight incubation with a rabbit polyclonal anti-Gβ1 antibody (Santa Cruz, CA, USA, C-16, 1:500) diluted in blocking solution at 4 °C. The membrane was washed 3 times with TBST and then probed with an anti-rabbit HRP-conjugated antibody for 1 h, and after 3 last washes, Gβ1 was revealed using a chemiluminescence kit (ECL select, Cytiva, Marlborough, MA, USA) and exposed in an imager (Amersham imager 600, Cytiva, Marlborough, MA, USA). Finally, to assure equal loading of protein, the membrane was stripped with 2 washes of guanidine HCl-based stripping solution (6 M GnHCl, 0.2% NP-40, 0.1 M β-mercaptoethanol in 20 mM Tris-HCl pH 7.5) followed by 3 washes of TBST and the house-keeping gene product β-tubulin was detected the same way with the corresponding primary (monoclonal mouse from Invitrogen, 1:10,000) and secondary antibodies (goat anti-mouse HRP-conjugated antibody from Sigma, St. Louis, MO, USA).

### 4.4. Cell Growth Assays

HEK 293 cells, as well as CRISPR/Cas9-generated ∆KCTD5, ∆KCTD2/5, and ∆KCTD2/5/17 lines, were cultured under standard conditions at 37 °C in a humidified 5% CO_2_ incubator. The culture medium used was DMEM (Invitrogen, Waltham, MA, USA), supplemented with 10% FBS and antibiotics (100 U/mL penicillin and 100 µg/mL streptomycin). For experimental procedures, cells were detached using trypsin and subsequently plated at a density of approximately ~20 × 10^3^ cells/cm^2^ in 6-well plates, allowing them to adhere and proliferate over a period of 54 h. Live cell imaging was performed on the plated cells using CELLCYTE X™ (CYTENA GmbH, Breisgau, Germany). Changes in confluency over time were quantified in the selected field on the plate, providing dynamic insights into cell behavior and growth patterns.

### 4.5. Calcium Measurements

An obelin biosensor was used as a calcium reporter, as described previously [37]. HEK 293 cells transfected with obelin-encoding plasmid were replated onto a white 96-well plate at the beginning of the experiment. They were then preincubated with 1 μM coelenterazine cp (Biotium, Fremont, CA, USA) at 37 °C in the dark for 2 h following washing. Subsequently, 1 mM carbachol was added, and total luminescence emission was monitored every 0.5 s for 30 s using a Tristar^2^ LB 942 Multimode Reader (Berthold, via MontrealBiotech, Montreal, QC, Canada). Results were expressed as normalized luminescence units (L.U.). The area under the curve (AUC) for 30 s of agonist stimulation was calculated and normalized to the maximal response induced by 0.1% Triton-X-100.

### 4.6. RNA Extraction, Library Preparation and Sequencing

RNA sequencing was performed as previously described [37,38,39]. Briefly, RNA from two independent samples of the parental and each of KCTD-KO cell lines was extracted using the RNeasy Mini Kit (Qiagen, Hilden, Germany) and homogenized using the QIAshredder (Qiagen, Hilden, Germany). The samples underwent quality control, poly-A library preparation, and RNA sequencing at the Genome Quebec Innovation Centre, following their standard protocol. Briefly, a Bioanalyzer 2100 (Agilent, Santa Clara, CA, USA) was used for quality control on each sample. First-strand cDNA was generated using random hexamer-primed reverse transcription, followed by second-strand cDNA synthesis using RNase H and DNA polymerase. Sequencing adapters were ligated using the TruSeq RNA Sample Preparation Kit (Illumina, San Diego, CA, USA). Fragments of approximately 350 bp were selected through gel electrophoresis, followed by 15 cycles of PCR amplification. The prepared libraries were then sequenced using an Illumina NovaSeq 2000 with four RNA-seq libraries per lane (paired-end 100 base pair).

### 4.7. Bioinformatic Analysis

The raw RNA-Seq data underwent quality control and adapter trimming using trimmomatic-0.39 to ensure the removal of low-quality bases and adapter sequences. Subsequently, the trimmed reads were aligned to the reference genome hg38 utilizing HISAT2 (Hierarchical Indexing for Spliced Alignment of Transcripts) with a pre-prepared indexed reference genome as previously described [37,38,39]. Aligned reads were then quantified for each gene using featureCounts, a tool designed for counting reads mapping to annotated features such as genes. The subsequent step involved differential expression analysis through DESeq2, a bioconductor package in R, encompassing the creation of a design matrix, statistical modeling, and identification of differentially expressed genes between conditions. The statistical significance of results was assessed using adjusted *p*-values, and differentially expressed genes were discerned based on log2fold changes. The results obtained from DESeq2 underwent comprehensive data interpretation, utilizing various visualization techniques, including a visually appealing heatmap (generated with pheatmap) as well as principal component analysis (PCA) to examine variability across samples and biological replicates. Additionally, KEGG pathway analysis was employed to elucidate the functional significance of differentially expressed genes. These integrated analyses contributed to a more thorough understanding of the underlying transcriptional landscape and its potential biological implications on cell growth and proliferation.

## Figures and Tables

**Figure 2 ijms-25-04993-f002:**
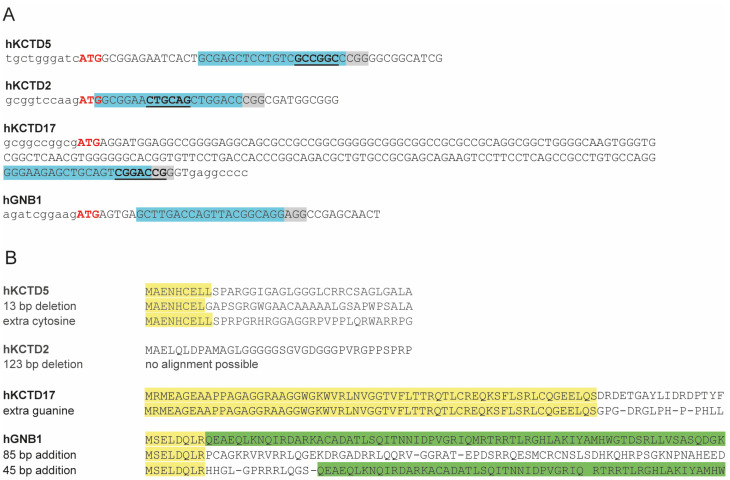
Generation of KCTD and GNB1 knockout lines in HEK 203 cells. (**A**) Localization of the sgRNA (in blue) relative to the Methionine start site (red); also indicated is the PAM sequence (grey) and the restriction site used for RFLP (bold and underlined); uppercase letters are coding sequence while non-coding is in lowercase letters. (**B**) Results of amplicon sequencing. Parental (in bold) and edited alleles (in normal) sequences were aligned using Cluster Omega (https://www.ebi.ac.uk/Tools/msa/clustalo (accessed multiple times in the last two years); yellow and green highlights indicate sequence homology to the parental sequence, a dash indicates a stop codon and the bold M (GNB1, 45 pb allele) indicates a possible alternative methionine translation start site.

**Figure 3 ijms-25-04993-f003:**
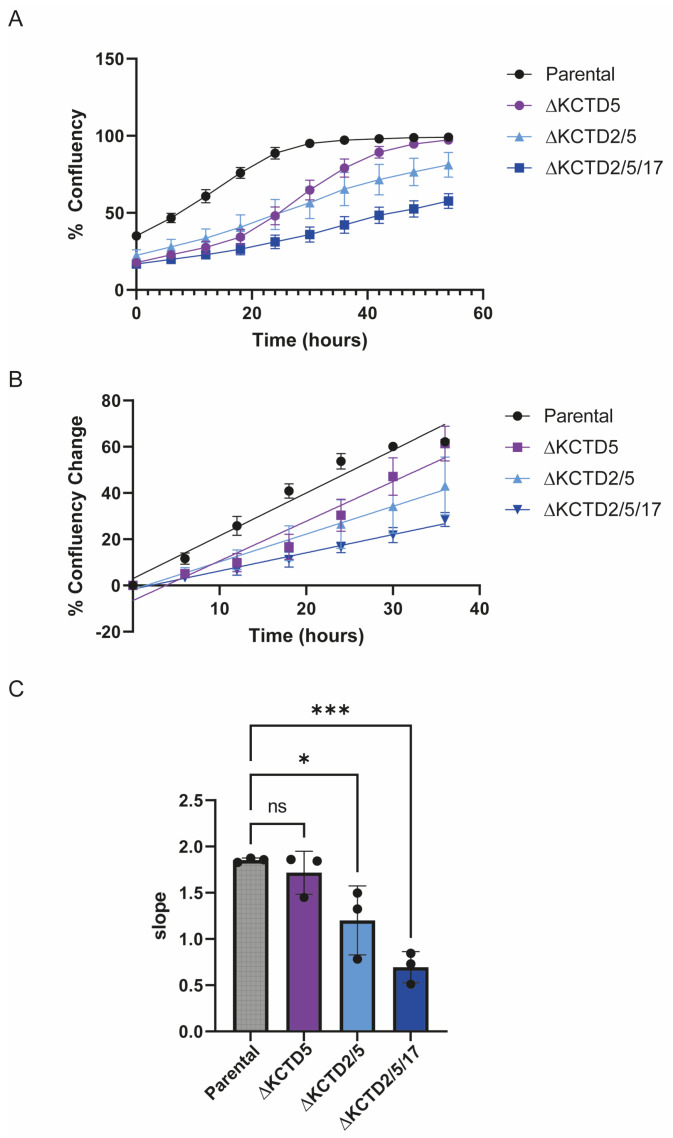
The effects of KCTD knockout on cell proliferation. Cell confluency curves for HEK 293 (parental), ∆KCTD5, ∆KCTD2/5, and ∆KCTD2/5/17 cells were measured and analyzed using a live cell imager (CELLCYTE). The percentage change in confluency was calculated as the ratio of confluency at each time point to the confluency at T = 0 h. (**B**) Slopes representing the % change in confluency from T = 0–36 h (where the curves in (**A**) were mostly linear) were calculated and shown in (**C**). The analysis was conducted based on an average of three independent experiments. Statistical significance was determined using one-way ANOVA, followed by Tukey’s multiple comparison test (ns, nonsignificant, * *p* < 0.05, *** *p* < 0.01). Error bars represent the standard error of the mean.

**Figure 4 ijms-25-04993-f004:**
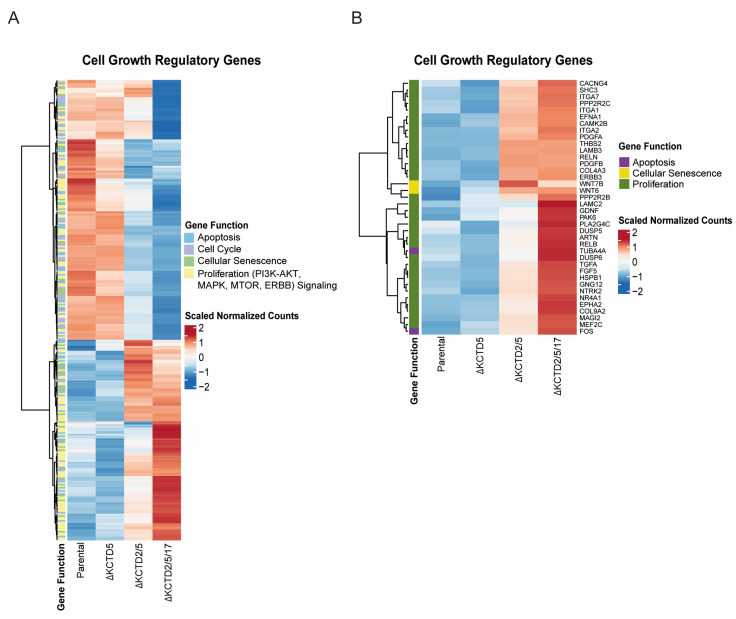
Changes in gene expression patterns related to cell growth phenotypes. (**A**) Heatmaps display data from RNA-seq experiments normalized scaled counts of differentially expressed genes linked to apoptosis, proliferation, cell cycle, and cellular senescence in KCTD knockout (KO) cell lines (∆KCTD5, ∆KCTD2/5, and ∆KCTD2/5/17). (**B**) 38 genes were found to have log2fold change ≥ 1.5, and base mean exceeding 100. Clustering reveals distinctive expression patterns, providing insights into the impact of progressive KCTD gene knockout on crucial cellular processes.

**Figure 5 ijms-25-04993-f005:**
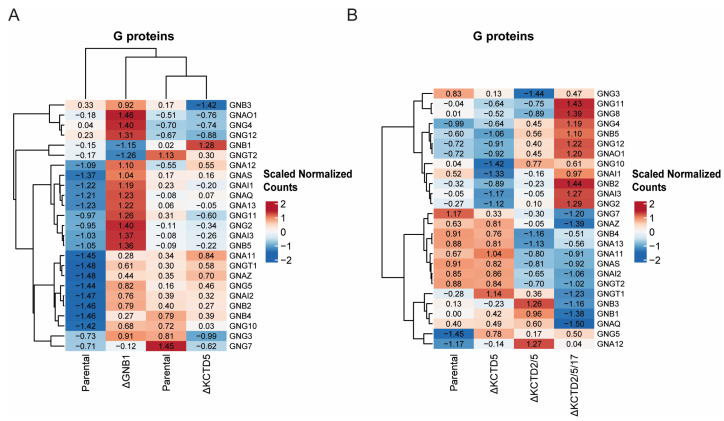
KCTD knockout effects on G protein expression. (**A**,**B**) Heatmaps depict normalized scaled counts of differentially expressed G protein genes in KCTD-KO cell lines (∆KCTD5, ∆KCTD2/5, ∆KCTD2/5/17) and GNB1-KO cell line. Clustering reveals distinct expression patterns.

**Figure 6 ijms-25-04993-f006:**
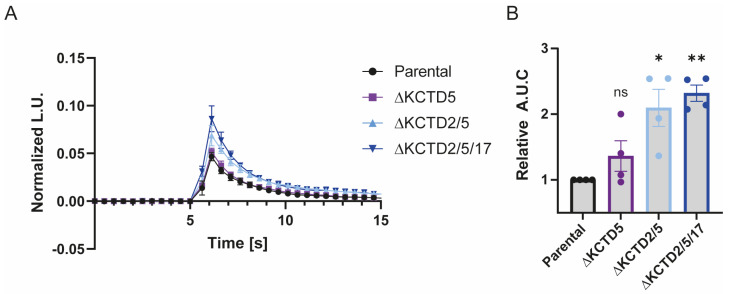
Calcium signaling is altered by KCTD knockout. HEK 293 cells endogenously expressing the M1/M3 subtypes of muscarinic acetylcholine receptors. Obelin, a calcium-sensing biosensor, was transfected into KCTD-KO cells as well as parental cells, which were loaded with 1 μM coelenterazine for 2 h and treated with 1 mM carbachol. Calcium release readings were obtained as a measure of luminescence emitted (L.U.), and values obtained were normalized as a percentage of maximum calcium release upon treatment with 0.1% Triton-x-100. (**A**) Calcium release kinetics, representative of 4 independent experiments, data are presented as mean ± S.E.M of three technical replicates. (**B**) Area under the Curve (A.U.C) was analyzed for each independent experiment and normalized to the parental line. Experimental analysis was performed from an average of four independent experiments with the one-sample *t*-test and Wilcoxon test (ns, nonsignificant, * *p* < 0.05, ** *p* < 0.01).

## Data Availability

The data are available at the Gene Expression Omnibus (series record GSE262642).

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
