# Peer review of "KCTD Proteins Have Redundant Functions in Controlling Cellular Growth"

_ijms, 2024, doi:10.3390/ijms25094993_

Round 1

Reviewer 1 Report

Comments and Suggestions for Authors

The manuscript entitled “KCTD proteins have redundant functions in controlling cellular growth.” provides new insights about the relationship between KCTD proteins and Gβ1 and a global role for this subfamily of KCTD proteins in maintaining the ability of cells to survive and proliferate. However, it needs major revision to make the suitable for publication.

1.      Authors should include information about choosing HEK293 cells compare to other cells.

2.      Also, include another cell lone model to avoid cell line specific results.

3.      What kind of cell death or cell cycle arrest after KO of KCTD?

Author Response

The manuscript entitled “KCTD proteins have redundant functions in controlling cellular growth.” provides new insights about the relationship between KCTD proteins and Gβ1 and a global role for this subfamily of KCTD proteins in maintaining the ability of cells to survive and proliferate. However, it needs major revision to make the suitable for publication.

  1. Authors should include information about choosing HEK293 cells compare to other cells.

Response: We chose HEK 293 cells because they are a well established cell model where we and others showed a robust interactions between Gbg subunits and KCTD5 via multiple mechanisms (Jiang et al., Sci Adv, 2023. 9(28): p. eadg8369, Sloan et al., J Biol Chem, 2023. 299(3): p. 102924, Campden et al., Methods Mol Biol, 2015. 1234: p. 161-84). KCTD proteins are ubiqitously expressed. We studied the effects of knocking out endogenous proteins, and showed that the effects could be rescued in a recently accepted manuscript. We have nuanced this in the discussion.

  1. Also, include another cell line model to avoid cell line specific results.

Response: In a recent manuscript, we showed that the purified versions of these proteins interact (Nguyen et al., in press PNAS), so cell type is not an issue. Previous studies have demonstrated the KCTD5 and Gbg subunits interact in striatal neurons as well (Muntean et al., 2022, referenced in our manuscript). We have changed the text to note this.

  1. What kind of cell death or cell cycle arrest after KO of KCTD?

Response: We have not yet explored this in detail and it is beyond the scope of the manuscript, but we have now left it as an open question in the revised discussion,

Reviewer 2 Report

Comments and Suggestions for Authors

The human Potassium (K+) Channel Tetramerization Domain (KCTD) family counts 25 members. Currently, their functional characterization is still ongoing, though their increasing relevance in various important biological functions is being gradually uncovered. Ever-growing evidence suggests that they might be implied in a broad range of functions including protein degradation. Additionally, emerging evidence suggests that several KCTD members might display potential roles either in cancer development or cancer prevention. Therefore, they represent appealing potential therapeutic targets in the treatment of tumors. In the manuscript titled "KCTD proteins have redundant functions in controlling cellular growth" Rizk R. and colleagues report the partial functional characterization (i.e., cell proliferation) of triple knock-out (KCTD2/5/17) cell line and the relationship with the Gb1 subunit by comparing the transcriptomic profile of the triple KO cell line and that of the Gb1 null.

Overall, the topic is interesting and the genetic approach used to dissect gene redundancy in the case of family members is appropriate.

Nonetheless, before publication in IJMS, quite a few flaws require to be amended. Shortly below are my concerns.

  1. I noticed that the authors submitted a manuscript to PNAS and the same manuscript has been deposited in the "biorxiv" repository, which "hosts" unpublished manuscripts. Since the manuscript has not been published I would strongly advise the authors to refer only to published materials. I warmly wish the authors all the best, including that the manuscript submitted will be accepted and soon released but by now seems to be still under revision, and in case the current manuscript is published earlier there would be a reference pointing out to a paper not released yet. Hence, please detail everything and skip referring to "biorxiv". This refers also to the asterisks present in Figure 1.
  2.   KO assessments require some amendments and some clarification. From Figure 2 and Figure S1 it is not clear whether both loci have been disrupted or just one. In other words, is KO in homozygosity? From Figure S1, the intensity of the band corresponding to the Gb1 in the KO lane is still present, though at a lower intensity letting surmise that a single locus has been disrupted. This issue has to be clarified by providing Western Blots, including the three KCTD members, and by genotyping properly the cell lines.
  3. Furthermore, controls of each single KO (i.e., DKCTD5 and DKCTD17) are required otherwise drawing solid conclusions on gene redundancy is quite tricky. Currently, in 293 cells we do not know whether DKCTD5, or DKCTD17, behave exactly like DKCTD2 in terms of proliferation, as well as Calcium signaling, or whether they might display a more severe, or milder, phenotype.
  4. Figure 3: since cell proliferation lasts up to 54 hrs, for consistency why not do the same with panels B and C (i.e., slope)?
  5. Figure 4 requires a better definition because even magnifying them it is hard to distinguish the gene name (Fig. 4B), whereas Fig. 5 seems a little bit messy. I would suggest to authors check it and at least cell line names should not overlap the heat map making it hard to distinguish among all of them.

Author Response

The human Potassium (K+) Channel Tetramerization Domain (KCTD) family counts 25 members. Currently, their functional characterization is still ongoing, though their increasing relevance in various important biological functions is being gradually uncovered. Ever-growing evidence suggests that they might be implied in a broad range of functions including protein degradation. Additionally, emerging evidence suggests that several KCTD members might display potential roles either in cancer development or cancer prevention. Therefore, they represent appealing potential therapeutic targets in the treatment of tumors. In the manuscript titled "KCTD proteins have redundant functions in controlling cellular growth" Rizk R. and colleagues report the partial functional characterization (i.e., cell proliferation) of triple knock-out (KCTD2/5/17) cell line and the relationship with the Gb1 subunit by comparing the transcriptomic profile of the triple KO cell line and that of the Gb1 null.

Overall, the topic is interesting and the genetic approach used to dissect gene redundancy in the case of family members is appropriate.

Nonetheless, before publication in IJMS, quite a few flaws require to be amended. Shortly below are my concerns.

  1. I noticed that the authors submitted a manuscript to PNAS and the same manuscript has been deposited in the "biorxiv" repository, which "hosts" unpublished manuscripts. Since the manuscript has not been published I would strongly advise the authors to refer only to published materials. I warmly wish the authors all the best, including that the manuscript submitted will be accepted and soon released but by now seems to be still under revision, and in case the current manuscript is published earlier there would be a reference pointing out to a paper not released yet. Hence, please detail everything and skip referring to "biorxiv". This refers also to the asterisks present in Figure 1.

Response: The PNAS paper is now accepted for publication, so we feel it adds value to this manuscript to refer to it. We will add the correct citation in the proofs if this paper is accepted.

  1. KO assessments require some amendments and some clarification. From Figure 2 and Figure S1 it is not clear whether both loci have been disrupted or just one. In other words, is KO in homozygosity? From Figure S1, the intensity of the band corresponding to the Gb1 in the KO lane is still present, though at a lower intensity letting surmise that a single locus has been disrupted. This issue has to be clarified by providing western blots, including the three KCTD members, and by genotyping properly the cell lines.

Response: If the reviewer is referring to KCTD17 KO genotyping results, an extra G addition is a common editing modification from the NHEJ DNA repair system so it is quite possible  that both alleles caried the same mutation. Also it is very unlikely that only one allele is mutated in this cell line since no WT allele was detected during sequencing. Regarding the western blot, in the Gb1 case, the lower bands is a non-specific band we have seen. Most G protein antibodies are not very good. There are no antibodies available for the KCTD isoforms but we have sequenced them and they are all gone.

  1. Furthermore, controls of each single KO (i.e., DKCTD5 and DKCTD17) are required otherwise drawing solid conclusions on gene redundancy is quite tricky. Currently, in 293 cells we do not know whether DKCTD5, or DKCTD17, behave exactly like DKCTD2 in terms of proliferation, as well as Calcium signaling, or whether they might display a more severe, or milder, phenotype.

Response: We suggest that the phenotype is related to redundancy as sequential deletion increases the effects. We have added a sentence considering the reviewer’s point. We are not (in a 2 week turn around period requested by the journal), going to generate several new KO lines, but we acknowledge this point in the revised discussion.

  1. Figure 3: since cell proliferation lasts up to 54 hrs, for consistency why not do the same with panels B and C (i.e., slope)?

Response: We only used the parts of the curves that were “linear” to calculate the slope. We have made that clearer in the Figure legend.

  1. Figure 4 requires a better definition because even magnifying them it is hard to distinguish the gene name (Fig. 4B), whereas Fig. 5 seems a little bit messy. I would suggest to authors check it and at least cell line names should not overlap the heat map making it hard to distinguish among all of them.

Response: The low-res versions for the review are likely responsible for this- we include hi-res versions for the journal to use.

Reviewer 3 Report

Comments and Suggestions for Authors

Comments and Suggestions for Authors

Robert Rizk,et al. in “KCTD proteins have redundant functions in controlling cellu-lar growth.”, showed a functional redundancy between the three isoforms, KCTD2, KCTD5 and KCTD17 , which has an important role in cell proliferation. They demonstrate that knocking out GNB1 exhibits contrasting effects compared to the knockout of KCTD isoforms regarding the modulation of G protein expression.

The article paper is well written and very original. It is an in-depth and very articulated study.

The authors should better explain:

- the rationship beetwen knocking out different KCTD isoform and the effects on calcium signaling.

The authors should add the concentration of the rabbit polyclonal anti-Gβ1 antibody and anti-beta-tubulin antibody in Immunodetection of Gβ1.

Comments on the Quality of English Language

Minor editing of English language required

Author Response

Robert Rizk,et al. in “KCTD proteins have redundant functions in controlling cellu-lar growth.”, showed a functional redundancy between the three isoforms, KCTD2, KCTD5 and KCTD17 , which has an important role in cell proliferation. They demonstrate that knocking out GNB1 exhibits contrasting effects compared to the knockout of KCTD isoforms regarding the modulation of G protein expression.

The article paper is well written and very original. It is an in-depth and very articulated study.

RESPONSE: Thanks for the kind words.

The authors should better explain:

- the relationship between knocking out different KCTD isoform and the effects on calcium signaling.

Response: The effects on calcium signalling are complicated. On the one hand knocking down Gb1 increases calcium signalling in HEK 293 cells, and knocking down Gb4 decreases it (Khan et al., 2015). Since KCTD proteins regulate both Gb isoforms, the complicated effect is likely a reflection of this. We mainly wanted to show that the effects become more striking during progressive knockdown of the three related KCTD isoforms. This is in the revised manuscript

The authors should add the concentration of the rabbit polyclonal anti-Gβ1 antibody and anti-beta-tubulin antibody in Immunodetection of Gβ1.

Response: We added these details in our revised Materials and Methods: rabbit polyclonal anti-Gβ1 antibody (1/500) and anti-beta-tubulin antibody (1/10000) in immunodetection of Gβ1.

Round 2

Reviewer 1 Report

Comments and Suggestions for Authors

Accept in current form.

Author Response

Thank you for your constructive comments.

Reviewer 2 Report

Comments and Suggestions for Authors

The revised version of the manuscript by Rizk R. et al titled "KCTD proteins have redundant functions in controlling cellular growth" has been slightly improved. However many issues remain open.

Point #1: the authors were asked to provide experimental evidence that was not provided. I understand that the authors might be very confident about their data, on the other hand, they also appear rather cautious in their reply (i.e.; "…it is quite possible that both alleles carried the same mutation. …", "…it is very unlikely that only one allele is mutated in this cell line since no WT allele was detected during sequencing"). Since it seems that discriminating homo- versus heterozygosity is an issue infeasible by using Ab, the authors are asked to provide, as supplementary materials, DNA-sequencing chromatograms to elucidate the issue, alongside, where possible, with PCR bands run on agarose gel to characterize the genomic DNA loci.

Regarding point #2, I understand that the standard time window for providing experimental evidence might be too short, but additional time can be asked to the Editor. I would suggest to the authors to ask for that and I am confident that the Editor, unambiguously, would be glad to offer some extra time to the authors.

Eventually, regarding the issue concerning figures 4 and 5, I might understand your reply concerning the low- versus high-definition/quality of the heatmap and the related gene names, but when it comes to the messy overlap (Figure 5) I am skeptical that this is related to the quality.

Author Response

COMMENT Point #1: the authors were asked to provide experimental evidence that was not provided. I understand that the authors might be very confident about their data, on the other hand, they also appear rather cautious in their reply (i.e.; "…it is quite possible that both alleles carried the same mutation. …", "…it is very unlikely that only one allele is mutated in this cell line since no WT allele was detected during sequencing"). Since it seems that discriminating homo- versus heterozygosity is an issue infeasible by using Ab, the authors are asked to provide, as supplementary materials, DNA-sequencing chromatograms to elucidate the issue, alongside, where possible, with PCR bands run on agarose gel to characterize the genomic DNA loci.

RESPONSE: We have now added a new Supplementary Figure which shows the sequencing results for all the KO lines. This was a good suggestion. Thanks.

COMMENT Regarding point #2, I understand that the standard time window for providing experimental evidence might be too short, but additional time can be asked to the Editor. I would suggest to the authors to ask for that and I am confident that the Editor, unambiguously, would be glad to offer some extra time to the authors.

RESPONSE: We acknowledge this a caveat in the Discussion. That said, to generate and characterize two new KO lines would take more than a year- in addition to the experiments to conduct in them. This is beyond the scope of the present manuscript.

COMMENT: Eventually, regarding the issue concerning figures 4 and 5, I might understand your reply concerning the low- versus high-definition/quality of the heatmap and the related gene names, but when it comes to the messy overlap (Figure 5) I am skeptical that this is related to the quality.

RESPONSE: Somehow in the generation of the version by the journal, Figure 5 was altered... we now include the correct image in the revised version. Also, we will send hi-res TIFF files for all figures! Thanks for point this out.

Round 3

Reviewer 2 Report

Comments and Suggestions for Authors

Though it is the third revision round it still seems that the manuscript is not ready yet for publication. Asking chromatogram was aimed to assess, based on the peak size, whether the mutation was homo- or heterozygous. Nonetheless, I appreciated the RFLP analysis. It seems that in Supplementary Figure 2C there is a mislabeling. Eventually, in the Discussion section, I missed the acknowledgment regarding the study's limitations, which in my opinion is essential. As such drawing solid conclusions on gene redundancy is a speculation and a misleading take-home message. Hence, the Discussion should be amended accordingly, by editing the part about redundancy, as well as the title.

Author Response

Thanks for your comments:

Regarding the chromatograms, this was not practical. As described in Materials and Methods, the nature of the mutations was determined by PCR amplicon sequencing from PCR products generated from genomic DNA coming from each KO cell line. These were cloned into a sequencing vector (pMiniT) and after bacterial transformation, plasmids were extracted from ~20 bacteria colonies were sent for Sanger sequencing. Sequencing results coming from this ganalysis was a sampling of individual DNA sequences representing the allele composition from each KO cell line and not a mixture of DNA sequences from which the chromatogram will display the relative composition of the nucleotide base(s) according to the relative size of the corresponding peak(s). This is the reason why we show the sequences rather than the chromatogram.

As for the mislabelling of Supplemental Figure 2, thanks for catching that. It has now been fixed.

Finally, we further revised the discussion to reflect the reviewer's concern. "Beyond this, it is clear that these three KCTD isoforms play additional roles that are so important that some redundancy needs to be built into their shared functions, which would need to be explored through generation of single KTCD2 and KCTD17 knockout lines. Further studies are required to assess redundancy directly but substituting each of the individual KCTD isoforms in the background of individual or double knockout lines. "